# Identifying multilevel predictors of behavioral outcomes like park use: A comparison of conditional and marginal modeling approaches

**Marilyn E. Wende** [1]*, **S. Morgan Hughey**[2], **Alexander C. McLain**[3], **Shirelle Hallum**[4], **J. Aaron Hipp**[5], **Jasper Schipperijn**[6], **Ellen W. Stowe**[4], **Andrew T. Kaczynski**[7]

1 Department of Health Education & Behavior, College of Health & Human Performance, University of Florida, Gainesville, FL, United States of America, 2 Department of Health and Human Performance, School of Health Sciences, College of Charleston, Charleston, SC, United States of America, 3 Department of Epidemiology and Biostatistics, Arnold School of Public Health, University of South Carolina, Columbia, SC, United States of America, 4 Department of Health Promotion, Education, and Behavior, Arnold School of Public Health, University of South Carolina, Columbia, SC, United States of America, 5 Department of Parks, Recreation, and Tourism Management, Center for Geospatial Analytics, North Carolina State University, Raleigh, NC, United States of America, 6 Department of Sports Science and Clinical Biomechanics, University of Southern Denmark, Odense, Denmark, 7 Department of Health Promotion, Education, and Behavior, Prevention Research Center, Arnold School of Public Health, University of South Carolina, Columbia, SC, United States of America

* Marilyn.wende@ufl.edu

**Data Availability Statement:** All relevant data are within the manuscript and its Supporting Information files.

## Abstract

This study compared marginal and conditional modeling approaches for identifying individual, park and neighborhood park use predictors. Data were derived from the ParkIndex study, which occurred in 128 block groups in Brooklyn (New York), Seattle (Washington), Raleigh (North Carolina), and Greenville (South Carolina). Survey respondents (n = 320) indicated parks within one half-mile of their block group used within the past month. Parks (n = 263) were audited using the Community Park Audit Tool. Measures were collected at the individual (park visitation, physical activity, sociodemographic characteristics), park (distance, quality, size), and block group (park count, population density, age structure, racial composition, walkability) levels. Generalized linear mixed models and generalized estimating equations were used. Ten-fold cross validation compared predictive performance of models. Conditional and marginal models identified common park use predictors: participant race, participant education, distance to parks, park quality, and population >65yrs. Additionally, the conditional mode identified park size as a park use predictor. The conditional model exhibited superior predictive value compared to the marginal model, and they exhibited similar generalizability. Future research should consider conditional and marginal approaches for analyzing health behavior data and employ cross-validation techniques to identify instances where marginal models display superior or comparable performance.

**Funding:** This study was supported by the National Cancer Institute (www.cancer.gov) under Award number R21CA202693 (PI: Andrew Kaczynski). The content is solely the responsibility of the authors and does not necessarily represent the official views of the National Institutes of Health. The funding agency did not play a role in the study design, data collection and analysis, decision to publish, or preparation of the manuscript.

**Competing interests:** The authors have declared that no competing interests exist.

## Introduction

Park use is associated with greater community cohesion [1,2], mental health [3], stress reduction [4,5], and physical activity (PA) [6,7]. Despite this, the percentage of the population that did not use parks increased from 25% to 30% from 1992 to 2015 [2]. To increase park use [8–10], it is important to identify key predictors of park use [11,12].

Methods for collecting information on park use predictors are diverse, including assessments of neighborhood, park, and individual-level characteristics [6,13–16]. At the neighborhood or administrative level, geographic information systems (GIS) tools are often used to map and quantify park availability [17], in conjunction with U.S. Census indicators (e.g., median household income) [18]. At the park level, audits conducted with reliable and valid tools are increasingly common [19], including the Community Park Audit Tool (CPAT) [20]. CPAT measures park access, neighborhood features, activity areas, quality, and safety [21,22]. At the individual level, intercept surveys administered to individuals visiting parks capture information about behaviors or sociodemographic characteristics [23,24]. While few studies sample park users *and* non-users [21,22,25–27], global positioning systems (GPS) and location-based resident surveys provide opportunities to study predictors of park use *and* non-use [22,28–36].

With more innovative and detailed methods for data collection, analyses that account for correlated observations are increasingly common. Observations can be clustered if there are multiple observations for the same individual (at several time points), *or* if data is hierarchically organized in the same geographic location (e.g., city, neighborhood) or organizational unit (e.g., school, workplace). In general, there are two popular types of statistical models used to account for correlated observations: conditional and marginal models [37,38]. For conditional models, clustered observations are accounted for through the inclusion of shared random effects in the linear predictor [39]. For marginal modeling, the mean function is modeled directly and the correlation structure is regarded as a nuisance parameter (i.e., it is not of interest) [37,38]. Conditional models specify a model on observation-level data, requiring assumptions on the random effect distribution and the independence of clustered observations given the random effects, which are often impossible to verify [37–40]. Alternatively, marginal methods model the average response on the population level and are robust to misspecification of the dependence structure [37,38]. Marginal models *also* make inferences about population (not individuals), which is advantageous for studying the impacts of multi-level factors on neighborhood park use.

In previous studies, analytical techniques that assume independent observations (e.g., ordinary least squares (OLS) regression) have been used to understand the relationship between multilevel factors and park use [8,21,22,41–44]. Loukaitou-Sideris and Sideris used OLS multiple linear regression to identify factors that bring children to parks, including recreation facilities, sport programs, natural features, and maintenance [42]. Additional studies have accounted for correlated observations, and largely used such methods to control for non-independence between people residing within the same geographic area using mixed regression, conditional models with random effects [45–52]. Stewart et al. linked GPS and individual travel diary data with park audit data and GIS measures for park access to examine if park facilities were related to the amount of park-based PA [53]. Authors used conditional models with random effects at the individual, park, and individual-park combination level [53].

Most past research on multi-level park use predictors employs conditional modeling. Limited research has compared conditional and marginal modeling techniques to determine advantages and disadvantages of different approaches for analyzing and interpreting multilevel park use predictors [38]. Therefore, this study will compare marginal and conditional

modeling techniques for identifying neighborhood, park, and individual level park use predictors. Methods from our study can be replicated to inform decision making on model selection for research on multilevel predictors of park use and other behavioral outcomes, and results can provide robust input on significant predictors of park use.

## Materials and methods

### Study design and setting

Data came from a parent study, ParkIndex, which is described fully elsewhere [21,54]. Data collection and recruitment occurred from Spring to Fall 2017 in four major U.S. localities: Brooklyn, NY, Greenville County, SC, Seattle, WA, & Raleigh, NC.

Census block groups (CBG) were classified into quartiles for park availability and income. CBGs in each location were enumerated by identifying all CBGs that fall within city and county limits (using Census shapefiles). Park availability and income quartiles were based on the number of parks intersecting the CBG and American Community Survey (ACS) 2011–2015 5-year estimates [18]. Using methods similar to other studies [45], 32 CBGs were selected within each location (128 total). This study included all public parks designed to facilitate active or passive use with an area size $\geq$ .25 acres. GIS park files for pre-selected CBGs were obtained from local parks and recreation agencies. A sample of residents in pre-selected CBGs were recruited to use an online, map-based survey (Maptionnaire.com) that included written consent (although consent was waived for this study) and displayed neighborhood parks to report park use *and* non-use and answer related questions [21]. Study procedures were granted exempt status by the University of South Carolina Institutional Review Board.

### Data collection

From June to October 2017, all parks in the study areas were audited in person by trained research staff using the CPAT [55]. CPAT contains four sections related to park information, access and surrounding area, activity areas, and quality/safety [55], and has demonstrated excellent inter-rater reliability and validity [46,56–59]. Overall, 275 parks were audited across the study areas: 94 in Seattle, 64 in Brooklyn, 71 in Raleigh, and 46 in Greenville County.

Adult residents within each CBG were recruited to take part in the study. Addresses in 32 CBGs per city were identified and mailed postcards containing a brief study description and online survey link. Postcards were mailed to verified residential addresses from June to October 2017, three separate times over a three-month period. Maptionnaire was used for administering the online survey [60]. Participants were shown a map of their neighborhood (i.e., area within a half-mile of the resident's CBG) and were asked to locate and answer questions about parks they had used within the past month, until the participant indicated they had not used any other parks in their neighborhood within the past month.

### Outcome measure

*Park visitation*. Participants were matched to parks within a ½-mile of their CBG and each of the person-park pairs was assigned yes/no based on reported park use in the past month.

### Explanatory measures–Neighborhood level

*Count of neighborhood parks*. The count of parks within a ½-mile (calculated by measuring the Network distance on ArcGIS 10.2.2) of each participant's CBG of residence.

*Median household income*. CBG-level data for the park addresses were downloaded from U.S. Census Bureau's ACS (5-year estimates, 2011–2015) [18]. Median household income was categorized into site-specific tertiles (low, intermediate, high).

*Population density*. CBG-level data for park addresses were downloaded from U.S. Census Bureau's ACS (5-year estimates, 2011–2015) [18]. Population density (population/area) was calculated and categorized into site-specific tertiles (low, intermediate, high).

*Age structure*. CBG-level data on the percentage of the population over the age of 65 years were downloaded from the U.S. Census Bureau's ACS (5-year estimates, 2011–2015) [18].

*Racial and ethnic composition*. CBG-level data for the park addresses were downloaded from the U.S. Census Bureau's ACS (5-year estimates, 2011–2015) [18], to calculate the percentage of the population identifying as non-Hispanic White.

*Walkability*. CBG-level National Walkability Index (1–20), based on the Environmental Protection Agency's Smart Location Mapping Database [61].

## Explanatory measures–Park level

*Distance to neighborhood parks*. Distance between participant addresses to each verified park entrance was calculated using Network distances on ArcGIS 10.2.2. Distances from residences to parks ranged from 0.01 to 12.7 miles (mean = 0.99, median = 0.82), since participants were asked to report park use within a half-mile buffer of their CBG of residence.

*Park quality*. Park quality (0–100) was calculated using CPAT average park access, facilities, amenities, quality concerns, aesthetics, and facility quality. For each component, scores were determined using yes/no responses to presence, usability and condition [21,22].

*Park size*. We calculated total acreage of each park, using GIS files for all selected parks.

## Explanatory measures–Participant level

*Physical activity*. Participants indicated the number of days and minutes per day spent engaged in moderate and vigorous PA in the past 7 days using the short form International PA Questionnaire [62]. Total number of weekly minutes of moderate and vigorous PA (MVPA) were summed for each participant [63], and categorized into low, intermediate, and high tertiles.

*Socio-demographics*. Respondents reported their race (White, American Indian, Alaskan native, Asian, Black or African American, Native Hawaiian, Pacific Islander, other), and gender (male, female, other). Age (in years) was calculated using the date of survey completion and reported date of birth. Race (White/Non-White) and gender (male/female) were dichotomized.

## Data management

A total of 360 residents completed the survey, and their data was merged with audit data for all parks within a ½-mile of their CBG of residence. The resultant long-form dataset included 3247 observations, representing all available neighborhood parks for each person. A total of 40 participants (relating to 157 observations) were removed from the analytic sample because they did not have any parks within a ½-mile of their CBG of residence or they only reported using parks outside the ½-mile CBG buffer, resulting in a final sample of 3090 observations and 320 participants. Each observation (n = 3090) included individual-level data (which was the same for all observations from the same participant), park-level data (i.e., audit data unique to each observation/park, reported park use), and CBG-level data (i.e., specific to each park address). On average, participants had 15 observations (i.e., 15 parks within the half-mile CBG buffer).

## Statistical analyses

**Descriptive statistics.** Descriptive information for participants included in this study were calculated and included ANOVA and chi-square tests to determine crude relationships between participant characteristics and park use. To account for participant non-response of survey data related to age (missing = 74), gender (missing = 45), PA (missing = 105), race and ethnicity (missing = 19), and education (missing = 48), hot deck single imputation was used (with PROC SURVEYIMPUTE, SAS 9.4). Hot deck imputation is a method for handling missing data in which each missing value is replaced with an observed response from a "similar" unit [64].

**Conditional model.** To evaluate park use predictors identified using varying statistical techniques, a two-level hierarchical conditional model was used with the imputed dataset. Level 1 was the participant-level and level 2 were the clusters of the units/participants nested within the same CBGs. The number of observations per participant related to the number of parks they could potentially report using in their CBG. Generalized Linear Mixed Models (GLMM) were used to examine the level 1 outcome (i.e., participant park use) as a function of predictors on level 1 and 2 using PROC GLIMMIX (SAS 9.4 software; see S1 File) with a quadrature estimation technique. Park-level clustering was not specifically accounted for, given that participants in the same CBGs were assigned to the same neighborhood parks. Participant's use of each park was treated as a binary outcome with a logit link function. Variance components were estimated for participant and neighborhood (i.e., CBG) level random effects. Models were presented after adjustment and included participant and CBG level random intercepts, fixed effects for different park visits, and individual level exposures. The equation below illustrates the conditional model predicting park use ($Y_{ij}$, binary outcome) of participant i in park j:

$$\text{Level 1}: \qquad Logit(P(Y_{ij} = 1)) = \beta_{0j} + \beta_1 X_{ij} + R_i$$

$$\text{Level 2}: \qquad \beta_{0j} = \gamma_0 + \gamma_1 Z_j + U_j$$

where $X_{ij}$ and $Z_j$ are the exposure variables, and $R_{ij}$ and $U_j$ are participant and CBG level random effects. Random effects are assumed to be normally distributed with $Var(R_{ij}) = \sigma^2$ and $Var(U_j) = \tau^2$. Model assumptions include: 1. distribution of the data conditional on the random effects is known, 2. probability of participant park use takes the form of a logistic mixed model, 3. random effects are normally distributed, 4. objective function for the optimization is a function of either the actual log likelihood, an approximation to the log likelihood, or the log likelihood of an approximated model [65]. Median Odds Ratio (MOR) quantified the amount of variance in the outcome for each level of the hierarchical model [66], using $MOR = \exp(\sqrt{2*V_a}*0.6745)$, where $V_a$ is the area level variance and 0.6745 is the 75th percentile of the cumulative distribution function of the normal distribution with mean 0 and variance 1 [66–68]. Before adjustment, CBG level MOR = 1.5 indicates that if a person were to move to a CBG with higher park use, median increase in the odds of park use (over all possible CBG) would increase by 50% [66]. Person level MOR = 1.5 indicate that if we compare all pairs of people in the same CBG, median OR, arranged such that the person with higher odds of park use is in the numerator, is 1.5 [66].

**Marginal model.** For analyzing marginal models for this study, a hierarchical structure with individuals nested within similar CBGs was used to estimate population level averages with the imputed dataset. As stated previously, the data for this study included multiple observations per participant, with different observations relating to unique neighborhood parks and their characteristics. Relationships between park use and participant and park-level exposure

variables were assessed using Generalized Estimating Equations (GEE) with a specified binary outcome distribution and a logit link function (i.e., PROC GEE, SAS 9.4 software; see S1 File). Repeated statements for CBGs and individuals nested within the same CBGs were included, and non-independence of the same parks used by different participants was accounted for on the CBG level. Quasi-likelihood-based criteria (QIC) values for the models with an independence correlation structure and with an exchangeable working correlation structure were compared to ensure independent correlation structure was best for this data. Model assumptions include: 1. the error terms for the same participant/geographic unit are correlated, 2. there is a linear relationship between the covariates and the transformation of the response (using logit link function, in this case), and 3. the covariance structure is correctly specified (GEE are robust to this last assumption). In addition, it is necessary that the sample size is sufficiently large (overall and within CBGs) for robust estimation of standard errors [38]. Models were presented after adjustment for different park visits and independent exposures. The equation below illustrates the marginal model predicting park use ($Y_{ij}$, binary outcome) of participant i in park j:

$$Logit(P(Y_{ij} = 1)) = \beta_0 X_0 + \beta_1 X_1 + \beta_1 t_1 + \beta_2 X_2 * \beta_2 t_2 + \beta_3 X_3 + \cdots + \beta_p X_p$$

where X are explanatory variables, and t refers to unique park visits for each person.

**Model comparisons.** We compared the direction of beta estimates for the same explanatory variables in full, adjusted GLMM and GEE models. We also compared which explanatory variables were significant predictors of park use in final (or reduced) GLMM and GEE models. Final, reduced models were chosen using backwards stepwise selection (based on α = 0.05 cutoff) and model fit. To assess model fit, penalized-likelihood information criteria were calculated for the GLMM model (i.e., Akaike & Bayesian Information Criterion), and quasi-likelihood-based criteria (i.e., QIC) were calculated for the GEE model. Marginal model odds ratios were interpreted as population averages, and the conditional model odds ratios were interpreted for the participant/CBG (i.e., holding all values fixed, including random effects).

The out of sample prediction error for each modeling approach were compared using tenfold cross validation methods [69]. Cross validation can assess the performance of predictive models to determine the generalizability of results [69], by randomly partitioning the original sample into 10 equal size subsamples. A single subsample is used as the validation data for testing the model, and the remaining 9 subsamples are used as training data. This process is repeated 10 times until each of the 10 subsamples are used once as validation data, and then the 10 results are averaged to produce a single estimation (S1 File) [69]. To compare predictive value using cross validation, we compared GLMM and GEE predictions based on fixed *and* random effects. To assess generalizability, we compared predictions based on *only* fixed effects. Data for this study can be found in S2 File.

## Results

### Sample characteristics

A total of 320 participants were included in the analytic sample, with 68.1% (n = 218) who reported using any neighborhood park(s) in the past month and 31.9% (n = 102) who did not (Table 1). Participants mostly reported intermediate PA levels (37.5%, n = 120), identified as White (69.1%, n = 221), were female (58.4%, n = 187), and had 2–4 years of college education (46.2%, n = 148). Average age of the sample was 47.1 years (SD = 15.3) (Table 1). Participants also had, on average, 57.4 park acres and 18.5 parks in their neighborhoods. These parks were 1.2 miles away and had a park quality score of 48.5, on average. Table 1 also shows the crude relationships between participant-level characteristics and park use. Results show that only

**Table 1. Sample characteristics and park use estimates for the study participants across four cities, N = 320.**

| | Total N (%) or Mean (SD) | Park user N (%) or Mean (SD) | Not a park-user N (%) or Mean (SD) | F/$\chi^2$ statistic (p-value) |
|---|---|---|---|---|
| **Total** | 320 (100) | 218 (68.1) | 102 (31.9) | - |
| **Physical activity level** | | | | |
| **High** | 95 (29.7) | 62 (28.4) | 33 (32.4) | .91 (0.6352) |
| **Intermediate** | 120 (37.5) | 81 (37.2) | 39 (38.2) | |
| Low[1] | 105 (32.8) | 75 (34.4) | 30 (29.4) | |
| **Race** | | | | |
| **Non-White** | 99 (30.9) | 74 (33.9) | 25 (24.5) | 2.9 (0.0888) |
| **White** | 221 (69.1) | 144 (66.1) | 77 (75.5) | |
| **Gender** | | | | |
| **Male** | 133 (41.6) | 95 (43.6) | 38 (37.2) | 1.1 (0.2848) |
| **Female** | 187 (58.4) | 123 (56.4) | 64 (62.8) | |
| **Other** | 0 (0%) | 0 (0%) | 0 (0%) | |
| **Age (in years)** | 47.1 (15.3) | 46.8 (15.9) | 47.7 (14.1) | 0.27 (0.6065) |
| **Education** | | | | |
| **Less than college** | 53 (16.6) | 41 (18.8) | 12 (11.8) | **12.4 (0.0021)** |
| **2–4 year college** | 148 (46.2) | 110 (50.5) | 38 (37.2) | |
| **Advanced degree** | 119 (37.2) | 67 (30.7) | 52 (51.0) | |
| **Study site** | | | | 4.8 (0.1845) |
| **Brooklyn, NY** | 46 (14.4) | 26 (11.9) | 20 (19.6) | |
| **Greenville County, SC** | 56 (17.5) | 37 (17.0) | 19 (18.6) | |
| **Raleigh, NC** | 82 (25.6) | 55 (25.2) | 27 (36.5) | |
| **Seattle, WA** | 136 (42.5) | 100 (45.9) | 36 (35.3) | |
| **Average size of neighborhood parks (in acres)** | 57.4 (443.6) | 66.1 (444.5) | 117.4 (748.8) | 0.2 (0.7020) |
| **Average distance to neighborhood parks (in miles)** | 1.2 (1.3) | 1.4 (1.6) | 0.9 (0.3) | **17.4 ($<$ .0001)** |
| **Average quality of neighborhood parks (out of 100)** | 48.5 (5.6) | 48.5 (5.4) | 48.5 (6.0) | 0.6 (0.4489) |
| **Count of neighborhood parks** | 18.5 (42.2) | 17.3 (41.0) | 23.1 (56.3) | 0.3 (0.5806) |

Bolded values are significant with α = .05

education level ($\chi^2$ = 12.4, p = 0.0021) and park distance (F = 17.4, p < .0001) had a significant relationship with park use.

## Model specifications

MOR estimates showed that if a person moves to another CBG with a higher probability of park use, their risk of park use will (in median) increase 3.1 times (MOR = 3.07). Block group level effects explained a large proportion of the variance in the outcome. This median increase in the OR is larger than the estimated OR for any predictor included in our model (see Table 2).

## Comparison of predictors of park use

Marginal and conditional models identified the same significant predictors of park use (and similar effect estimates) in the full model, aside from park size (Table 2 and S1 Fig). Significant predictors of park use in the reduced, underline{conditional model} included participant race, participant education, distance to parks, park size, park quality, and percent of the neighborhood population age 65 and older (Table 2). For the participant level, there were 43% decreased odds of

**Table 2. Predictors of participant park use identified using conditional and marginal models, N = 3090.**

| | | Conditional (GLMM) Full Model | | Conditional (GLMM) Reduced Model | | Marginal (GEE) Full Model | | Marginal (GEE) Reduced Model | |
|---|---|---|---|---|---|---|---|---|---|
| | | Odds Ratio (95% Confidence Interval) | p-value | Odds Ratio (95% Confidence Interval) | p-value | Odds Ratio (95% Confidence Interval) | p-value | Odds Ratio (95% Confidence Interval) | p-value |
| **Participant** | Physical activity level | | | | | | | | |
| | High | 1.17 (0.81, 1.69) | 0.3958 | | | 1.02 (0.72, 1.43) | 0.9282 | | |
| | Intermediate | 0.99 (0.68, 1.44) | 0.9637 | | | 1.21 (0.87, 1.68) | 0.2593 | | |
| | Low[1] | - | - | | | - | - | | |
| | Race | | | | | | | | |
| | White | - | - | - | - | - | - | - | - |
| | Non-White | 0.58 (0.40, 0.82) | **0.0025** | 0.57 (0.41, 0.79) | **0.0009** | 0.61 (0.43, 0.87) | **0.0058** | 0.66 (0.48, 0.91) | **0.0115** |
| | Education | | | | | | | | |
| | Less than college | - | - | - | - | - | - | - | - |
| | 2–4 year college | 1.47 (0.90, 2.40) | 0.1247 | 1.90 (1.20, 3.01) | **0.0061** | 1.59 (1.08, 2.34) | **0.0178** | 2.04 (1.40, 2.97) | **0.0002** |
| | Advanced degree | 1.89 (1.13, 3.15) | **0.0154** | 2.49 (1.55, 3.98) | **0.0002** | 1.97 (1.33, 2.93) | **0.0007** | 2.58 (1.77, 3.74) | **< .0001** |
| | Gender | | | | | | | | |
| | Male | 0.92 (0.68, 1.26) | 0.6063 | | | 0.91 (0.69, 1.21) | 0.5280 | | |
| | Female | - | - | | | - | - | | |
| | Age (in years) | 0.99 (0.98, 1.00) | **0.0199** | | | 0.99 (0.98, 1.00) | **0.0132** | | |
| **Park** | Distance to parks | 1.75 (1.48, 2.08) | **< .0001** | 1.78 (1.55, 2.05) | **< .0001** | 1.83 (1.48, 2.27) | **< .0001** | 1.95 (1.67, 2.29) | **< .0001** |
| | Park size | 1.00 (1.00, 1.00) | **0.0376** | 1.00 (1.00, 1.00) | **0.0142** | 1.00 (1.00, 1.00) | 0.2487 | | |
| | Park quality score | 1.01 (1.00, 1.02) | 0.1354 | 1.02 (1.01, 1.03) | **< .0001** | 1.01 (1.00, 1.02) | **0.1612** | 1.02 (1.01, 1.03) | **0.0002** |
| **Neighborhood** | Median household income | | | | | | | | |
| | High | 1.83 (0.97, 3.43) | 0.0611 | | | 1.82 (1.00, 3.31) | 0.0492 | | |
| | Intermediate | 1.25 (0.77, 2.02) | 0.3601 | | | 1.25 (0.81, 1.93) | 0.3201 | | |
| | Low[1] | - | - | | | - | - | | |
| | Percent Non-Hispanic White | 1.00 (0.99, 1.00) | 0.2727 | | | 1.00 (0.99, 1.00) | 0.2083 | | |
| | Population density | | | | | | | | |
| | High[1] | - | - | | | - | - | | |
| | Intermediate | 1.15 (0.72, 1.82) | 0.5584 | | | 1.07 (0.71, 1.62) | 0.7532 | | |
| | Low | 1.20 (0.72, 1.98) | 0.4893 | | | 1.21 (0.78, 1.88) | 0.3902 | | |
| | Percent of population > 65yr | 1.04 (1.02, 1.06) | **0.0004** | 1.02 (1.01, 1.03) | **< .0001** | 1.03 (1.01, 1.05) | **0.0007** | 1.02 (1.01, 1.03) | **< .0001** |
| | National Walkability Index | 1.01 (0.95, 1.07) | 0.6929 | | | 1.02 (0.96, 1.08) | 0.5065 | | |
| | Count of parks | 0.97 (0.95, 1.00) | 0.0224 | | | 0.98 (0.96, 1.00) | 0.0435 | | |
| **Model fit statistics** | AIC/BIC | 2800.08/2850.87 | | 3244.95/3271.59 | | - | | - | |
| | QIC | - | | - | | 2193.81 | | 2463.15 | |
| **Ten-fold cross validation** | RMSE (fixed & random effects) | 0.32 | | 0.32 | | 0.35 | | 0.36 | |
| | MAPE (fixed & random effects) | 0.21 | | 0.22 | | 0.52 | | 0.52 | |
| | RMSE (fixed effects) | 0.35 | | 0.36 | | 0.35 | | 0.36 | |
| | MAPE (fixed effects) | 0.52 | | 0.52 | | 0.52 | | 0.52 | |

Akaike Information Criterion (AIC)

Bayesian Information Criterion (BIC)

Root Mean Squared Error (RMSE) & Mean Absolute Percentage Error (MAPE)

[1]Referent value for categorical predictors

park use for Non-White compared to White participants, and 90% increased odds of park use for those with 2–4 year college and 149% increased odds of park use for those with an advanced degree compared to less than college (holding other predictors fixed). For the park level, there were 78% increased odds of park use for every mile distance to neighborhood parks, 0.07% increased odds of park use for every one unit increase in park acreage, and 2% increased odds of park use for every one unit increase in park quality for each given participant (holding other predictors fixed). For the neighborhood level, there were 2% increased odds of park use for every percent increase in residents over 65 years (holding other predictors fixed, including random effects).

Significant predictors of park use in the reduced, <u>marginal model</u> included participant race, participant education, distance to parks, park quality, and percent of the neighborhood population age 65 and older (Table 2). For the participant level, on average, there were 34% decreased odds of park use for Non-White compared to White participants, 104% increased odds of park use for those with 2–4 year college, and 158% increased odds of park use for those with an advanced degree compared to less than college. For the park level, on average, there was 95% increased odds of park use for every mile distance to parks and there were 2% increased odds of park use for every one unit increase in park quality. For the neighborhood level, on average, there were 2% greater odds of park use for every one percent increase in residents over 65 years.

Conducting tenfold cross validation with both fixed and random effects for the conditional model, Root Mean Squared Error (RMSE) for the conditional model was 0.32 compared to 0.36 for the marginal model (Table 2). Mean Absolute Percentage Error (MAPE) for the conditional model was 0.22, compared to 0.52 for the marginal model (Table 2). RMSE and MAPE were lower for the conditional model, meaning conditional models had superior <u>predictive value</u> of results. With *only* fixed effects for the conditional model, RMSE for the conditional model was 0.36 compared to 0.36 for the marginal model. MAPE for the conditional model was 0.52, compared to 0.52 for the marginal model (Table 2). RMSE and MAPE were similar between conditional and marginal models, so <u>generalizability</u> was comparable.

## Discussion

Conditional and marginal models identified similar park use predictors on the individual- and park-levels. Both models identified participant race, participant education, distance to each park from residence, park quality, and percentage of the neighborhood population over 65 years as park use predictors. Additionally, the conditional model identified park acreage as a predictor, although the effects of park acreage on park use in the conditional model were modest. Results from ten-fold cross validation showed that the conditional model exhibited superior predictive value compared to the marginal model, but they exhibited similar generalizability.

Previous research that used conditional models have reported that number of nearby parks [46], neighborhood park space [46], park proximity [46,50], neighborhood walkability [48,49], park quality [46,50], and violent crime were park use predictors [52]. Few studies indicated that White and higher income or educated groups were more likely to report park use compared to racial minority populations and lower income or educated populations [46,48]. Our results showed that increased distance to neighborhood parks related to higher odds of park use. This contradicts past research showing that perceived park proximity positively impacts park use [48], and other research establishing a null association between objectively-measured proximity and park use [70]. This finding could in part be explained by the fact that participants were only able to report parks used within a ½-mile of their CBG of residence (ranging

0.01 to 12.7 miles). Even the furthest parks were still somewhat proximal to participant residences. Though logic may indicate that a close park would result in greater visitation, growing evidence also shows that park use and park-based PA is influenced by a variety of complex factors, including facilities present, quality, social interaction, safety, and demographic characteristics [21,22,41–43,45–49,71–74]. Further, our sample was predominately White and educated, which may indicate greater access to resources that influence travel patterns to parks, such as car transportation [75–77].

Findings showing conditional models had superior predictive value compared to marginal may partially validate the use of conditional modeling in past research [45–52]. Cross validation results also demonstrated that generalizability is similar between marginal and conditional models when only comparing predicted values using fixed effects. Past research has compared conditional and marginal approaches [78–80], and stated that marginal (GEE) models may be favorable since they are robust to misspecification of the covariance structure [78,79]. Conditional (GLMM) models only provide unbiased interpretation when the correct random effects has been selected [78], so the significance of the factors that contribute to the different behavioral outcomes are more reliable from marginal models [78,79]. Our research introduced ten-fold cross validation as a method for assessing predictive value and generalizability of these different statistical approaches that can be replicated [69]. For interpretations, the conditional model slope coefficients reflect the average *within neighborhood parks* log(OR) of park use and the marginal model log(OR) reflects probabilities of park use *averaged across all neighborhood parks*, both as a result of significant neighborhood, park, and participant level predictors [38]. The goal of this research was to estimate park use predictors across all neighborhood parks, so the marginal model may be preferable for interpretation purposes.

This study has important research implications. First, we carefully outlined modeling approaches for handling multi-level predictors for park use, which can be applied to a wide range of health behavior outcomes that are influenced by environmental/structural factors. Researchers can use similar procedures as those outlined in this paper to compare analytical approaches for complex datasets and conduct more appropriate statistical analyses. As methods for collecting data on multiple levels of influence become more advanced and multi-level interventions gain popularity, commentary on related analytic approaches is imperative [12,81–84]. Second, our study provides input on methods for multi-level data collection. Researchers should consider available tools for multi-level data collection as they plan their research and incorporate more distal influences of health behaviors (e.g., policies, neighborhoods), such as those outlined in social ecological models [11,12]. Third, our study used imputation methods to account for missing data on the individual level and minimize bias [85]. Future research can employ more rigorous techniques by incorporating follow up phone calls or engaging communities in data collection to establish commitment/trust [85–88]. Lastly, researchers can build on our findings by collecting multi-level data longitudinally to understand if there is a temporal relationship between identified predictors and park use. Longitudinal research can employ GEE weighting methods to account for missing data over time (i.e., drop outs, skipped time points) [89,90].

Findings from this research can also be used to ameliorate low park use and promote outdoor PA [91,92]. Specifically, determining neighborhood-, park-, and individual-level park use predictors in diverse geographic locations was a significant practice-based contribution of this study. In addition, methods used to collect multi-level data on park use predictors in this study can be employed by local public health practitioners to better understand community-level resource access. For instance, Maptionnaire.com surveys for individual-level, CPAT for park-level, and Census Bureau's ACS data for neighborhood-level data were carefully chosen by our study team as accessible, high-quality options for multi-level measurement [15,18,60].

This research had several limitations. While this study offered granular-level data on park use patterns for adults in four geographically diverse U.S. cities, the survey sample was more affluent and had higher percent female than the target population and park users may be more interested and willing to answer a survey regarding local parks, which may have skewed our sample/estimates. Conversely, ~68% of participants in our study reported being park users, which closely matches past estimates of park use [93]. Next, some variables may be subject to over and under-estimation, such as self-reported PA. While this research employed a valid and reliable questionnaire [94], findings may reflect this potential bias. Finally, despite repeated recruitment efforts, the survey sample was smaller than anticipated with only about 300 participants and 3,000 potential park visits. Since our study used mail-based recruitment and had low response rates, future research should adopt novel approaches for community recruitment, such as social media-based recruitment which has showed promise for location-based sampling and recruiting during COVID-19 [95,96].

## Conclusions

This study takes an empirical approach to addressing low levels of park use by comparing methods for analyzing multi-level predictors of park visitation, including both well-known conditional and more underutilized marginal approaches. Conditional and marginal modeling techniques identified similar park use predictors and showed similar generalizability, and conditional models showed superior predictive value. This is the first study to compare these popular analytical approaches to determine multi-level predictors of park use, and calls attention to underused marginal modeling approaches and its strengths when handling multi-level data. Future research can intentionally compare key features of conditional and marginal approaches for analyzing park use and other health behavior data, including their assumptions, interpretations, and specific applications outlined in this paper. Adding to this, researchers can employ cross validation techniques to identify instances where marginal models may display superior or comparable predictive performance.

## Supporting information

**S1 Checklist. Human participants research checklist.**
(DOCX)

**S1 File. Sample code for conditional models, marginal models, and ten fold cross validation.**
(DOCX)

**S2 File. ParkIndex dataset used for analyses.**
(XLSX)

**S1 Fig. Odds ratios with 95% confidence limits for conditional and marginal models predicting park use.**
(TIF)

## Author Contributions

**Conceptualization:** Marilyn E. Wende, S. Morgan Hughey.

**Data curation:** Marilyn E. Wende, S. Morgan Hughey, Shirelle Hallum, J. Aaron Hipp, Jasper Schipperijn, Ellen W. Stowe, Andrew T. Kaczynski.

**Formal analysis:** Marilyn E. Wende, S. Morgan Hughey, Alexander C. McLain, Andrew T. Kaczynski.

**Funding acquisition:** Andrew T. Kaczynski.

**Methodology:** Marilyn E. Wende, S. Morgan Hughey, Shirelle Hallum, J. Aaron Hipp, Jasper Schipperijn, Ellen W. Stowe.

**Project administration:** Andrew T. Kaczynski.

**Resources:** Andrew T. Kaczynski.

**Software:** Andrew T. Kaczynski.

**Supervision:** Andrew T. Kaczynski.

**Visualization:** Marilyn E. Wende.

**Writing – original draft:** Marilyn E. Wende, S. Morgan Hughey.

**Writing – review & editing:** Marilyn E. Wende, S. Morgan Hughey, Alexander C. McLain, Shirelle Hallum, J. Aaron Hipp, Jasper Schipperijn, Ellen W. Stowe, Andrew T. Kaczynski.

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
