## [Decision Letter · Decision Letter 0]

14 Feb 2024

PONE-D-23-37844Identifying multilevel predictors of behavioral outcomes like park use: A comparison of conditional and marginal modeling approachesPLOS ONE

Dear Dr. Wende,

Thank you for submitting your manuscript to PLOS ONE. After careful consideration, we feel that it has merit but does not fully meet PLOS ONE’s publication criteria as it currently stands. Therefore, we invite you to submit a revised version of the manuscript that addresses the points raised during the review process.

We look forward to receiving your revised manuscript.

Kind regards,

Sathishkumar Veerappampalayam Easwaramoorthy

Academic Editor

PLOS ONE

Journal Requirements:

This study was supported by the National Cancer Institute (www.cancer.gov) under Award number R21CA202693 (PI: Andrew Kaczynski). The content is solely the responsibility of the authors and does not necessarily represent the official views of the National Institutes of Health. The funding agency did not play a role in the study design, data collection and analysis, decision to publish, or preparation of the manuscript.

3. In the online submission form, you indicated that Data from the ParkIndex study will be made available upon request, submitted via email to the corresponding author. Data cannot be shared publicly because home addresses or location data is included in this dataset, but can be de-identified or removed upon request for specific research application. SAS code for this research is provided as a supplemental file with this manuscript.

Reviewers' comments:

Reviewer's Responses to Questions

**Comments to the Author**

1. Is the manuscript technically sound, and do the data support the conclusions?

Reviewer #1: Yes

Reviewer #2: Partly

2. Has the statistical analysis been performed appropriately and rigorously? 

Reviewer #1: Yes

Reviewer #2: Yes

3. Have the authors made all data underlying the findings in their manuscript fully available?

Reviewer #1: No

Reviewer #2: Yes

4. Is the manuscript presented in an intelligible fashion and written in standard English?

Reviewer #1: Yes

Reviewer #2: Yes

5. Review Comments to the Author

Reviewer #1: Comments to Authors,

This observational study looks at the predictors of park use, combining data from park characteristics, census block groups, and individual survey data. This use of multi-level data, the comparison of conditional and marginal modelling approaches, and the application of cross validation make this an interesting study that would have value to the scientific community. Additionally, the authors laid out the assumptions for each model which is valuable.

The two weaknesses in the study that I found were both beyond the control of the authors. First, the study is observational rather than causal, so the usual caveats about observational studies apply. However, observational studies are still very valuable. Second, the authors do not have data regarding children in the household which I believe would be an important predictor of park use. These issues do not change my view that the study is publishable.

Suggestions or things to consider:

1. Abstract sentence 1 and throughout the rest of the paper I suggest that you change “marginal and conditional approaches” to “marginal and conditional modeling approaches” which is how you state it in the title and several other places.

2. Explanatory Measures – Participant Level page 6. Was gender dichotomized because the only responses were male/female or is that how it appeared on the survey? If it appears on the survey that way, then that may explain some of the missing gender responses. Gender is not dichotomous. Nothing to do here, but something to consider for future research.

3. Explanatory Measures. Is there an omitted variable problem since there is no data on whether or not a respondent has school age children? This seems like a determinant of park use. Given the data, I understand that it cannot be controlled for at this point. Does it merit comment?

4. Statistical Analysis page 7. I am fine with the hot deck imputation, but later in the Discussion (page 3) you state that the imputation helps minimize bias. Are the results very different without the imputation?

5. Sample Characteristics section. Could you add some park descriptive statistics?

6. Sample characteristics page 11. The sample is skewed very female. Does this deserve comment?

7. Model Specifications page 11. Can you add to this single sentence section? Can you provide context for the size of this effect? It would be helpful if this were explained more.

8. Discussion page 2. “The goal of this research … so the marginal model may be preferable for interpretation purposes.” Should this also be mentioned earlier such as in the introduction or where you are describing the two types of models?

Minor Comments:

9. In the abstract, it would be easier to follow for the reader if you just stated that the marginal model found the same predictors as the conditional model, except for park size.

10. Introduction page 1, “In general, there are two types …” to “In general, there are two popular types … “ since there are other possible approaches.

11. Introduction page 2. Typo – “used such methods control” to “used such methods to control”

12. Explanatory Measures page 4. Typo change “eac” to “each”.

13. Data Collection page 4. The last sentence says “This sequence was repeated until … “ What, specifically does the sequence refer to? The entire postcard sequence or just the identification of parks visited?

14. page 9. I am not a big fan of stepwise selection since it highlights sample idiosyncrasies, however I am fine with it since you also present the full model results. Do not cut out the full model results.

15. Sample characteristics page 10. Table 1 is in an odd place, but you will fix that in publication.

16. Discussion page 1. “… park acreage on park use in the conditional model were marginal.” Please choose a different word than marginal here since we don’t want to confuse it with the marginal model.

17. Discussion page 1. “… proximity and park use.33” I don’t know what the 33 is for.

Reviewer #2: The authors have studied the marginal and the conditional approaches for identifying individual, park, and neighborhood park use predictors. There are some collected results in this study however there are some drawbacks that the authors should address them to improve this study.

1.In the introduction, the main contribution of this study must be provided to outline the prioritization of this comparative work.

2.The proposed models about multilevel predictors of behavioral outcomes must be provided in this study.

3.The description of dataset must be indicated and presented through the distribution areas as well as the specific database.

4.The collected results must be presented through the distributive data figures.

5.The comparative methods must be implemented in the discussion.

6.The novelty points of this study must be indicated clearly in the conclusion part.

6. PLOS authors have the option to publish the peer review history of their article (what does this mean?). If published, this will include your full peer review and any attached files.

Reviewer #1: No

Reviewer #2: No

---

## [Author Response · Author response to Decision Letter 0]

12 Mar 2024

Response to Review

Manuscript: PONE-D-23-37844

Title: Identifying multilevel predictors of behavioral outcomes like park use: A comparison of conditional and marginal modeling approaches

Thank you to the reviewers for their helpful feedback on our manuscript. The original reviewer comments are listed below followed by our responses preceded by “Response:”. All major changes have been highlighted in the text of the revised paper. 

Reviewer #1: 

This observational study looks at the predictors of park use, combining data from park characteristics, census block groups, and individual survey data. This use of multi-level data, the comparison of conditional and marginal modelling approaches, and the application of cross validation make this an interesting study that would have value to the scientific community. Additionally, the authors laid out the assumptions for each model which is valuable.

Response: Thank you for your positive remarks. 

The two weaknesses in the study that I found were both beyond the control of the authors. First, the study is observational rather than causal, so the usual caveats about observational studies apply. However, observational studies are still very valuable. Second, the authors do not have data regarding children in the household which I believe would be an important predictor of park use. These issues do not change my view that the study is publishable.

Response: Thank you for your analysis of the weaknesses of this research. While we agree that additional research including prospective data collection and the participation of children is important, it is outside the scope of the current research (as you’ve pointed out).

Suggestions or things to consider:

1. Abstract sentence 1 and throughout the rest of the paper I suggest that you change “marginal and conditional approaches” to “marginal and conditional modeling approaches” which is how you state it in the title and several other places.

Response: Thank you for this comment. We have updated this wording in the abstract (page 

iii, lines 3-4) and throughout the manuscript. 

2. Explanatory Measures – Participant Level page 6. Was gender dichotomized because the only responses were male/female or is that how it appeared on the survey? If it appears on the survey that way, then that may explain some of the missing gender responses. Gender is not dichotomous. Nothing to do here, but something to consider for future research.

Response: Thank you for this comment. We have explained in the methods (page 6, line 5) that “other” was included as an option. That said, no respondents chose this category so it was not included in the tables. To clarify, we have included “other” in Table 1 with no responses. 

3. Explanatory Measures. Is there an omitted variable problem since there is no data on whether or not a respondent has school age children? This seems like a determinant of park use. Given the data, I understand that it cannot be controlled for at this point. Does it merit comment?

Response: Unfortunately, this question was not included for our online survey. We did ask participants who they visited each park with but given that question was specific to each park and had a high level of missingness we chose to exclude it from the current analysis. That said, we agree that additional consideration should be made about living with school-aged children in future research on this topic. 

4. Statistical Analysis page 7. I am fine with the hot deck imputation, but later in the Discussion (page 3) you state that the imputation helps minimize bias. Are the results very different without the imputation?

Response: Although results were similar using imputed and non-imputed datasets, imputation was used to reduce selection bias due to non-response and ensure we had the power to compare the two models under study accurately. Generally, imputation is used to minimize bias in these forms, so we think it is wise to use imputed data for this analysis. 

5. Sample Characteristics section. Could you add some park descriptive statistics?

Response: We have added park characteristics to page 10, lines 10-15 and in Table 1. 

6. Sample characteristics page 11. The sample is skewed very female. Does this deserve comment?

Response: We have included this as a limitation in this study, as we agree that readers should consider this when interpreting results (see page 18, lines 21-23 to page 19, lines 1-2). 

7. Model Specifications page 11. Can you add to this single sentence section? Can you provide context for the size of this effect? It would be helpful if this were explained more.

Response: We have added additional explanation on page 11, lines 4-6. 

8. Discussion page 2. “The goal of this research … so the marginal model may be preferable for interpretation purposes.” Should this also be mentioned earlier such as in the introduction or where you are describing the two types of models?

Response: This has been previously mentioned on page 2, lines 7-8. 

Minor Comments:

9. In the abstract, it would be easier to follow for the reader if you just stated that the marginal model found the same predictors as the conditional model, except for park size.

Response: Thank you for this suggestion. We have updated the wording on page iii, lines 14-17. 

10. Introduction page 1, “In general, there are two types …” to “In general, there are two popular types … “ since there are other possible approaches.

Response: We have added this clarification to page 1, lines 22-23.

11. Introduction page 2. Typo – “used such methods control” to “used such methods to control”

Response: We have added this correction to page 2, line 14.

12. Explanatory Measures page 4. Typo change “eac” to “each”.

Response: We have added this correction to page 4, line 20.

13. Data Collection page 4. The last sentence says “This sequence was repeated until … “ What, specifically does the sequence refer to? The entire postcard sequence or just the identification of parks visited?

Response: The sequence referred to identifying parks they used and answering questions about them, not the entire postcard process. We have changed the wording on page 4, lines 11-14 to clarify this. 

14. page 9. I am not a big fan of stepwise selection since it highlights sample idiosyncrasies, however I am fine with it since you also present the full model results. Do not cut out the full model results.

Response: We agree that it is important to include both full and reduced model results, which are presented in Table 2. 

15. Sample characteristics page 10. Table 1 is in an odd place, but you will fix that in publication.

Response: We agree that this must be adjusted for final publication. 

16. Discussion page 1. “… park acreage on park use in the conditional model were marginal.” Please choose a different word than marginal here since we don’t want to confuse it with the marginal model.

Response: Thank you for this comment. We agree that the wording should be changed from “marginal” to “modest” on page 16, line 6. 

17. Discussion page 1. “… proximity and park use.33” I don’t know what the 33 is for.

Response: We have corrected this error on page 16, line 17. 

Reviewer #2: 

The authors have studied the marginal and the conditional approaches for identifying individual, park, and neighborhood park use predictors. There are some collected results in this study however there are some drawbacks that the authors should address them to improve this study.

1.In the introduction, the main contribution of this study must be provided to outline the prioritization of this comparative work.

Response: Thank you for this comment. We have provided more detail on the main contribution of this study on page 3, lines 1-4.

2.The proposed models about multilevel predictors of behavioral outcomes must be provided in this study.

Response: Response: Details on the models being compared in this study are provided in the Statistical Analysis section, on page 7 line 4 to page 10 line 3. 

3.The description of dataset must be indicated and presented through the distribution areas as well as the specific database.

Response: We have now provided the data for readers to access, and a description of the dataset has been provided on page 6, lines 7-17. We have also added information on the distribution of participants across study sites in Table 1. 

4.The collected results must be presented through the distributive data figures.

Response: We have included data figures as supplemental file 2, which is a figure that represents the results presented in Table 2. If we did not interpret this comment correctly, we would happily make additional revisions. 

5.The comparative methods must be implemented in the discussion.

Response: We have provided an analysis of our findings and how it relates to past research on page 16, lines 1-23 to page 17, lines 1-18. This also includes narration on the importance of the ten-fold cross-validation methods we used to compare the two modeling approaches. If the reviewer is looking for a specific change related to the discussion of comparative methods in the discussion, we would be happy to consider it. 

6.The novelty points of this study must be indicated clearly in the conclusion part.

Response: We have added commentary on the novelty of this study on page 19, lines 16-18. Thank you for your comment.

In conclusion, thank you to the reviewers for providing constructive feedback on this research. With these changes, we believe we have responded to and addressed all the identified concerns and that the updated manuscript has been improved substantially due to our collective efforts. If further revisions are required, we would be happy to consider them. Again, thank you for the opportunity to resubmit our manuscript to PLOS ONE and we look forward to hearing back from you soon.

---

## [Decision Letter · Decision Letter 1]

18 Mar 2024

Identifying multilevel predictors of behavioral outcomes like park use: A comparison of conditional and marginal modeling approaches

PONE-D-23-37844R1

Dear Dr. Wende,

We’re pleased to inform you that your manuscript has been judged scientifically suitable for publication and will be formally accepted for publication once it meets all outstanding technical requirements.

Kind regards,

Sathishkumar Veerappampalayam Easwaramoorthy

Academic Editor

PLOS ONE

Additional Editor Comments (optional):

Reviewers' comments:

Reviewer's Responses to Questions

**Comments to the Author**

1. If the authors have adequately addressed your comments raised in a previous round of review and you feel that this manuscript is now acceptable for publication, you may indicate that here to bypass the “Comments to the Author” section, enter your conflict of interest statement in the “Confidential to Editor” section, and submit your "Accept" recommendation.

Reviewer #1: All comments have been addressed

Reviewer #2: (No Response)

2. Is the manuscript technically sound, and do the data support the conclusions?

Reviewer #1: Yes

Reviewer #2: (No Response)

3. Has the statistical analysis been performed appropriately and rigorously? 

Reviewer #1: Yes

Reviewer #2: (No Response)

4. Have the authors made all data underlying the findings in their manuscript fully available?

Reviewer #1: Yes

Reviewer #2: (No Response)

5. Is the manuscript presented in an intelligible fashion and written in standard English?

Reviewer #1: Yes

Reviewer #2: (No Response)

6. Review Comments to the Author

Reviewer #1: (No Response)

Reviewer #2: (No Response)

7. PLOS authors have the option to publish the peer review history of their article (what does this mean?). If published, this will include your full peer review and any attached files.

Reviewer #1: No

Reviewer #2: No

---

## [Editor Report · Acceptance letter]

4 Apr 2024

PONE-D-23-37844R1 

PLOS ONE

Dear Dr. Wende, 

I'm pleased to inform you that your manuscript has been deemed suitable for publication in PLOS ONE. Congratulations! Your manuscript is now being handed over to our production team.

Kind regards, 

on behalf of

Dr. Sathishkumar Veerappampalayam Easwaramoorthy 

Academic Editor

PLOS ONE